# *Mangifera indica* L. Leaves as a Potential Food Source of Phenolic Compounds with Biological Activity

**DOI:** 10.3390/antiox11071313

**Published:** 2022-06-30

**Authors:** Giuseppe Sferrazzo, Rosa Palmeri, Cristina Restuccia, Lucia Parafati, Laura Siracusa, Mariarita Spampinato, Giuseppe Carota, Alfio Distefano, Michelino Di Rosa, Barbara Tomasello, Angelita Costantino, Massimo Gulisano, Giovanni Li Volti, Ignazio Barbagallo

**Affiliations:** 1Department of Drug and Health Sciences, University of Catania, Viale A. Doria 6, 95125 Catania, Italy; giuseppesferrazzo95@gmail.com (G.S.); barbara.tomasello@unict.it (B.T.); angelita.costantino@unict.it (A.C.); massimo.gulisano@unict.it (M.G.); 2Department of Agricultural, Food and Environment, University of Catania, 95123 Catania, Italy; rosa.palmeri@unict.it (R.P.); cristina.restuccia@unict.it (C.R.); lucia.parafati@unict.it (L.P.); 3Istituto di Chimica Biomolecolare del CNR (ICB-CNR), Via Paolo Gaifami 18, 95126 Catania, Italy; laura.siracusa@icb.cnr.it; 4Department of Biomedical and Biotechnological Sciences, University of Catania, Via S. Sofia 87, 95125 Catania, Italy; mariarita.spampinato@phd.unict.it (M.S.); giuseppe.carota@phd.unict.it (G.C.); alfio.distefano@unict.it (A.D.); mdirosa@unict.it (M.D.R.); livolti@unict.it (G.L.V.); 5Interuniversity Consortium for Biotechnology, Area di Ricerca, Padriciano, 34149 Trieste, Italy

**Keywords:** *Mangifera indica* L., bioactive compound, antimicrobial, antioxidant, antifibrotic

## Abstract

It is well recognized that functional foods rich in antioxidants and antiinflammation agents including polyphenols, probiotics/prebiotics, and bioactive compounds have been found to have positive effects on the aging process. In particular, fruits play an important role in regular diet, promoting good health and longevity. In this study, we investigated on biological properties of extract obtained from *Mangifera indica* L. leaves in preclinical in vitro models. Specifically, the profile and content of bioactive compounds, the antimicrobial potential toward food spoilage and pathogenic bacterial species, and the eventually protective effect in inflammation were examined. Our findings revealed that MLE was rich in polyphenols, showing a content exclusively in the subclass of benzophenone/xanthone metabolites, and these phytochemical compounds demonstrated the highest antioxidant capacity and greatest in vitro antibacterial activity toward different bacterial species such as *Bacillus cereus*, *B. subtilis*, *Pseudomonas fluorescens*, *Staphylococcus aureus*, and *St. haemolyticus*. Furthermore, our data showed an in vitro anti-inflammatory, antioxidant, and antifibrotic activity.

## 1. Introduction

In the human body, biochemical compounds present in our diet, i.e., antioxidants, work toward the maintenance of health and vigor, especially for healthy aging [1]. These are well known to be good for health and can be obtained through tea, vegetable, and fruit intake. In particular, phenolic and carotenoid compounds in leafy vegetables or fruits have excellent antioxidant properties in scavenging free radicals [2,3,4]. In this context, *Mangifera indica* L. (mango) has proven to be a good example.

Mango is a stone fruit belonging to the Anacardiaceae family grown in many parts of the world, particularly in tropical areas. It is appreciated for its nutritional value, attributable also to the presence of health-enhancing compounds; it is considered a good source of ascorbic acid, carbohydrates, dietary fiber, carotenoids, organic acids, and phenolic compounds [5]. In particular, several studies have shown that mango extract possesses anticancer activity [6] and plays an anti-inflammatory function in numerous chronic pathological disorders associated with an inflammatory response [7,8]. In our previous study, we evaluated the protective effect of mango leaf extract (MLE) on murine adipose tissue inflammation, validating that the phytochemical compounds of mango leaves show functional activity toward adipocytes, improving lipid metabolism and functionality [9]. In recent years, the interest in plant extracts, particularly those obtained from agro-industry byproducts (peels, leaves, and seeds) has grown considerably, not only in the nutraceutical field, but also in food preservation [10,11,12]. For this reason, the control of pathogenic microorganisms and/or food spoilage throughout the food chain contributes considerably to the reduction of pre- and post-production losses and is important for human health. In this context, the antimicrobial activity of mango leaves has been documented in several studies. Indeed, extracts obtained using different solvents, such as water, ethanol, methanol, chloroform, benzene, and acetone, have shown the ability, albeit with different efficacy levels, to inhibit the growth of Gram-positive and -negative bacteria [13,14,15], but fewer studies have documented the antifungal activity of MLE [16]. Here, we investigated the biological properties of an extract obtained from Sicilian mango leaves, Kensington pride cultivar. In particular, the profile and the content of bioactive compounds in the leaf extract, the antimicrobial potential against bacterial species pathogenic for humans and food, and the possible cytotoxicity and protective effect on in vitro preclinical models of inflammation were analyzed.

## 2. Materials and Methods

### 2.1. Chemicals

All reagents and solvents used in this study were of analytical grade and without further purification. Pure reference standards gallic acid, mangiferin, rutin (quercetin 3-*O*-rutinoside), and benzophenone were bought from Sigma Aldrich (Milan, Italy); HPLC solvent-grade water and acetonitrile were from VWR (Milan, Italy). LX-2 and U937 human cell lines were purchased from ATCC (Milan, Italy).

### 2.2. Preparation of the Extract

The experimental analysis was conducted on mango leaves of the Kensington pride cultivar, obtained from Sicilian mango (Lentini, Italy). Leaves were separately dried in a stove at 36 °C until constant weight, and then crushed with a pestle and mortar until obtaining a homogeneous sample. Each sample was subjected to hydroalcoholic extraction. The extract was obtained according to Yi Zhang et al.’s [17] method, with some modifications. Briefly, the samples were extracted in 70% ethanol (*v*/*v*) for 30 min and subsequently filtered by a vacuum pump. Aliquots of the resulting extracts were immediately analyzed to assess profile and content of bioactive compounds in order to evaluate the corresponding antioxidant activity, while the remaining part was freeze-dried and used for the other tests/assays/determinations.

### 2.3. HPLC/DAD and HPLC/ESI-MS Analyses

Hydroalcoholic extracts from red mango leaves were analyzed as described below. Chromatographic analyses were performed using an Ultimate3000 UHPLC system equipped with a high-pressure binary pump, a photodiode array detector, a thermostatted column compartment, and an automated sample injector (Thermo Fisher Scientific, Inc., Milan, Italy). The collected data were processed through a Chromeleon v. 6.80 Chromatographic information management system. The chromatographic analyses were performed using a reversed-phase column (Gemini C18, 250 × 4.6 mm, particle size 5 μm, Phenomenex Italia s.r.l., Bologna, Italy) equipped with a guard column (Gemini C18 4 × 3, 0 mm, particle size 5 μm, Phenomenex Italia s.r.l., Bologna, Italy). Mango leaf polyphenols were eluted using a gradient of B (2.5% formic acid in acetonitrile) in A (2.5% formic acid in water): 0 min, 5% B; 10 min, 15% B; 30 min, 25% B; 35 min, 30% B; 50 min, 90% B; 57 min then kept for other 7 min, 100% B. The solvent flow rate was 1 mL/min, the temperature was kept at 25 °C, and the injector volume setting was 10 μL. Quantification was performed at 280 nm for gallic acid and its derivatives using gallic acid (*R*^2^ = 0.9998) as standard; the same wavelength was adopted to build the calibration curve for maclurin and iriflophenones using benzophenone (*R*^2^ = 0.9998) as the external reference. Mangiferin was quantified at 330 nm using the corresponding commercial reference (*R*^2^ = 0.9999), while rutin (*R*^2^ = 0.9998) was used to quantify flavonols in the extract; in this latter case, quantification was carried out at 350 nm. In order to unambiguously identify the chromatographic signals and/or to confirm peak assignments, additional HPLC/ESI-MS analyses were performed. The HPLC apparatus was the same described above, and ESI mass spectra were acquired by a Thermo Scientific Exactive Plu Orbitra MS (Thermo Fisher Scientific, Inc., Milan, Italy), using a heated electrospray ionization (HESI II) interface. Mass spectra were recorded operating in negative ion mode in the *m**/z* range 120–1500 at a resolving power of 25,000 (full width at half maximum, at *m/z* 200, RFWHM), resulting in a scan rate >1.5 scans/s when using an automatic gain control target of 1.0 × 10^6^ and a C-trap inject time of 250 ms. under the following conditions: capillary temperature 300 °C; nebulizer gas (nitrogen) with a flow rate of 60 arbitrary units; auxiliary gas flow rate of 10 arbitrary units; source voltage 3 kV; capillary voltage 82.5 V; tube lens voltage 85 V. The Orbitrap MS system was tuned and calibrated in positive mode by infusion of solutions of a standard mixture of sodium dodecyl sulfate (Mr 265.17 Da), sodium taurocholate (Mr 514.42 Da), and Ultramark (Mr 1621 Da). Data acquisition and analyses were performed using the Excalibur software. All analyses were performed in triplicate.

### 2.4. Antioxidant Activity Measurement

The hydroalcoholic extracts from red mango leaves were filtered with a 0.45 µm pore size membrane filter (Millipore^®^, Burlington, MA, USA) and analyzed for their antioxidant activity using the DPPH radical-scavenging activity method as previously reported by Brand-Williams et al. (1995) [18], with small adjustments. Briefly, 100 μM of 1,1-diphenyl-2-picrylhydrazyl (DPPH) (Sigma-Aldrich, Milan, Italy) was dissolved in methanol (Sigma-Aldrich, Milan, Italy), and 3 mL of this solution was mixed with 50 μL of each extract. The mangiferin standard was dissolved in ethanol to obtain a final concentration of 2.02 mg/mL equal to that present in the MLE, and each suspension was also subjected to the DPPH assay. All samples were incubated for 60 min at room temperature; then, the absorbance decrease at 515 nm (AE) was measured by spectrophotometry. DPPH radicals have a maximum absorption at 515 nm; the peak disappears following reduction caused by an antioxidant compound. A white sample was used as a reference, containing 70 μL of methanol solution in the DPPH solution. The data obtained were used to calculate the percentage of DPPH inhibition that was expressed as radical-scavenging activity (RSA) and calculated as follows:RSA (%) = ((absorbance blank − absorbance sample)/absorbance blank) × 100.

### 2.5. In Vitro Evaluation of Antibacterial Activity of Mango Leaf Extract

The freeze-dried extracts of mango leaves were re-suspended in a fixed volume of dimethyl sulfoxide (DMSO) and subsequently sterilized by filtration using a 0.20 μm pore size syringe filter (Millipore^®®^, Burlington, MA, USA), whereupon the antibacterial activity of each extract was evaluated. In particular, the following selected food pathogenic and spoilage bacteria belonging to the collection of the Department of Agricultural, Food, and Environment (Di3A) in Catania were investigated: *Bacillus cereus*, *B. subtilis*, *Escherichia coli*, *Listeria gray*, *L. monocytogenes*, *Pseudomonas fluorescens*, *Salmonella enterica*, *Staphylococcus aureus*, and *St. hemolyticus*.

The antimicrobial activity against the aforementioned bacterial species was also evaluated for mangiferin standard, which was suspended in DMSO at the same concentration of extracts, i.e., 2.02 mg/mL.

Bacterial strains were grown in Nutrient Broth (NB; Biolife Italiana S.r.l., Milano, Italy) at 37 °C for 24 h. After incubation, each strain was diluted with sterile Ringer’s solution to obtain a final concentration of 10^6^ cells/mL; subsequently, each bacterial suspension was individually inoculated in Petri plates containing Nutrient Agar (NA; Biolife Italiana S.r.l., Milano, Italy). Wells were made, using a sterile cork borer (5 mm diameter), on the surface of the inoculated NA plate and then filled with 60 μL of extract. The same volume of sterile DMSO was added in the reference negative control well. The inhibitory effect of the extracts against the targeted microbial strains was assessed after 24–48 h of incubation at 24 °C, by measuring the size (cm) of the inhibition zone (no microbial growth) around the well. Results were expressed as the mean ± standard deviation of two biological replicates, and each replicate was evaluated as a triplicate.

### 2.6. Measurement of Cell Viability: MTT Assay

Cell cultures were treated for 24 h in 96-well plates with different concentrations of MLE (35 μg/mL, 75 μg/mL, and 150 μg/mL), and the medium was subsequently substituted by a solution containing bromide 3-(4,5-dimethylthiazol-2-yl)-2,5-diphenyltetrazolium (MTT) and incubated for 3 h at 37 °C. Finally, 100 μL of dimethyl sulfoxide (DMSO) was added to each well, and the absorbance was detected using a spectrophotometer at λ = 570 nm.

### 2.7. Alpha-Glucosidase Inhibition Assay

Alpha-glucosidase (α-glucosidase) from *Saccharomyces cerevisiae* (E.C. 3.2.1.20) was prepared in potassium phosphate (0.1 mol/L, 3.2 mmol/L MgCl_2_, pH 6.8) using *p*-nitrophenyl-α-d-glucopyranoside as the substrate for the reaction, at a final concentration of 6 mmol/L. The assay reaction was prepared by mixing 282 μL of MLE with 200 μL of substrate, incubated at 37 °C for 5 min; after the incubation, 200 μL of enzyme was added to the solution. The enzyme reaction was performed at 37 °C for 15 min and stopped by 1.2 mL of glycine buffer (pH 10). The enzyme activity was measured spectrophotometrically by determining absorbance at 410 nm. In order to measure the inhibitory effect, the activity was compared with a control using water. Data obtained were expressed as the inhibition percentage and calculated as follows:Inhibition (%) = (ΔA_410_ control − ΔA_410_ extract)/ΔA_410_ control × 100.

Enzymatic inhibition was expressed as the IC_50_ value, calculated using a linear plot of the inhibition curve, in accordance with the following equation:*y* = −0.496*x* + 93.489.

### 2.8. Anti-Inflammatory Activity In Vitro

U937 and LX-2 cell lines were used to measure the anti-inflammatory activity of MLE. U937 cells are human promonocytic myeloid leukemia cells isolated from a histiocytic lymphoma. This cell line shows many monocyte-like aspects and is widely used in the literature. The cells were differentiated into macrophages using200 nM PMA (phorbol 12-myristate 13-acetate) for 72 h. After incubation, the suspended cells were removed by aspiration, and the adherent cells (U937Φ) were washed three times with 1× PBS (phosphate-buffered saline). In order to simulate the inflammation, the cells were further incubated with or without LPS at the concentration of 100 ng/mL for 2 h with the same medium (RPMI 1960, 10% FBS). After the LPS treatment, MLE was added at 35 μg/mL for 4 h [9]. Cells were divided into four groups of treatment as follows: U937Φ, U937Φ + LPS, U937Φ + MLE, and U937Φ + LPS + MLE. After 6 h of treatment, cells were collected by trypsinization, washed once with 1× PBS, and then lysed for RNA extraction.

Moreover, LX-2, a cell line of human hepatic stellate cells, was used to confirm the anti-inflammatory ability and additionally investigate the antifibrotic activity of extracts. LX-2 cells were seeded in growth medium Dulbecco’s modified Eagle’s medium (DMEM) with 1 g/L d-glucose (Gibco by life technologies, Milan, Italy), supplemented with 10% *v/v* heat-inactivated fetal bovine serum (FBS; Invitrogen) and 1% penicillin/streptomycin (Carlo Erba) antibiotic/antimycotic solution. The cultures were maintained at 37 °C in a 5% CO_2_ incubator, and the medium was replaced after 48 h and every 3–4 days thereafter. To simulate the inflammation, cells were further incubated with or without LPS at the concentration of 100 ng/mL for 2 h with the same medium (DMEM with 1 g/L d-glucose). After the LPS treatment, MLE was added at 35 μg/mL for 4 h [9]

### 2.9. RNA Extraction and qRT-PCR

Total RNA extraction was obtained using Trizol reagent (Life Technology, Milan, Italy). Extracted RNA was converted into cDNA using an Applied Biosystem (Foster City, CA, USA) reverse transcription kit. The quantitative analysis was carried out with the One-Step Fast Real-Time PCR System Applied Biosystem using the SYBR Green PCR master mix (Life Technology, Milan, Italy). The primer sequences used are shown in Table 1. The PCR analysis mix contained synthesized cDNA, SYBR Green PCR master mix, primer mix (forward primer/reverse primer), and UltraPureTM DNase/RNase-free distilled water (Invitrogen by Life Technologies, Milan, Italy). PCR reactions were subjected to 40 cycles of 95 °C for 20 s, 95 °C for 3 s, and 60 °C for 30 s. The relative mRNA expression levels of genes were determined according to the threshold cycle (Ct) value of each PCR product and normalized with GAPDH (glyceraldeyde-3-phosphate dehydrogenase) as the housekeeping gene using the comparative 2^−ΔΔCt^ method.

### 2.10. Statistical Analysis

The statistical significance (*p* < 0.05) of the differences between the experimental groups was determined by single-factor analysis of variance (ANOVA) using Dunnett’s test for the analysis of multiple comparisons. Data are presented as the mean ± SD.

## 3. Results

### 3.1. Polyphenol Profile and Content of MLE

The HPLC/DAD chromatograms of mango leaf extracts, visualized at 280 nm, are shown in Figure 1. Fifteen peaks were tentatively recognized on the basis of their relative retention times, UV/Vis and mass spectral data, injection with pure analytical standards, and comparison with literature data. In the chromatogram of red mango leaves, polyphenols were the most represented class of metabolites, particularly the subclass of benzophenone/xanthone derivatives [19]. In fact, six peaks over 15 (peaks 2, 3, 4, 6, 7, and 8) corresponded to maclurin and iriflophenone derivatives, while mangiferin, the peculiar xanthone of this species, was present with one metabolite only (peak 5). Peaks 1, 11, 14, and 15 corresponded to gallic acid (peak 1) and its derivatives with a hexose, plausibly glucose. Flavonols, mainly quercetin derivatives (peaks 9, 10, 12, and 13), were the only subclass of flavonoids found in the extract. Iriflophenone 3-C-glucoside (peak 3 in Figure 1) was by far the main compound with 4.218 mg/mL in the extract coming from red leaves, followed by mangiferin (peak 5 in Figure 1; 2.023 mg/mL in red leaf extract; see also Table 2). Red mango leaves are rich in polyphenols, plausibly due to a higher accumulation of defense metabolites in the earlier stage of leaf development [20]. Interestingly, in the case of mango leaves, overaccumulation seemed to affect almost exclusively the subclass of benzophenone/xanthone metabolites, whilst the other subclasses did not undergo remarkable changes (Table 2). Our data showed an exclusive high concentration of iriflophenone 3-C-glucoside and mangiferin contained in the leaves (Figure 1).

### 3.2. Antioxidant Activity of Mango Leaf Extracts

The antioxidant activity of MLE is displayed in Table 3. The extract obtained from leaves evidenced the highest RSA of 90.19% ± 0.31%, followed by the standard solution containing 2.02 mg/mL of mangiferin (87.80% ± 1.86% RSA).

### 3.3. Antibacterial Activity of Mango Leaf Extract

Results obtained from the agar well diffusion assay, performed on NA medium, evidenced the antimicrobial efficacy of MLE, depending on the bacterial strain. Specifically, data reported in Table 4 prove that the extract possessed antimicrobial activity. The MLE showed the significantly (*p* < 0.05) highest antimicrobial activity against *Ps. fluorescens*, followed by *St. haemolyticus* and *St. aureus*, producing inhibition halo values of 0.70 ± 0.00, 0.50 ± 0.00, and 0.43 ± 0.06 cm, respectively. Although with a lower width of inhibition halos, encouraging results were also obtained against *B. cereus* and *B. subtilis* strains (0.43 ± 0.06 and 0.23 ± 0.06 cm, respectively). MLE did not show any inhibitory effect against *E. coli*, *L. gray*, *L. monocytogenes*, and *S. enterica*. Standard mangiferin solution at the concentration of 2.02 mg/mL, equal to the mangiferin concentration found in MLE (see Table 4), did not exert antibacterial activity against any target bacteria.

### 3.4. Effect of MLE on Cell Viability

The in vitro cytotoxic activity of MLE was investigated in two susceptible human cell lines, human hepatic stellate cells (LX-2) and macrophages (U937). Figure 2 shows that the tested concentrations of MLE exhibited no significant differences in cell viability, indicating no in vitro cytotoxic activity toward the two cell lines analyzed.

### 3.5. Alpha-Glucosidase Inhibition

As shown in Figure 3, the mango leaf extract dose-dependently inhibited α-glucosidase according to the plot of percentage α-glucosidase activity vs. concentration (µg/mL) of MLE. The mango leaf extract exhibited significant α-glucosidase-inhibitory activity with an IC_50_ of 187.48 μg/mL.

### 3.6. MLE Reduces In Vitro Inflammation and Hepatic Fibrosis

Our data showed a higher antioxidant and antibacterial profile of the leaf extract; therefore, its possible bioactive effect on in vitro models of inflammation was subsequently analyzed. U937 cells, differentiated into macrophages, were used as an in vitro model to evaluate the anti-inflammatory activity of MLE. Differentiated macrophages were treated with lipopolysaccharide (LPS), and, after 2 h, the MLE was added at 35 μg/mL for 4 h. The expression of proinflammatory genes such as cyclooxygenase-2 (COX-2) (involved in the production of prostaglandins) [21], interleukin-6 (IL6), interleukin-1β (IL-1β), and tumor necrosis factor-α (TNF-α) was evaluated. Figure 4 shows that, in macrophages treated with LPS, there was an increase in proinflammatory genes compared to the control group. The cotreatment with MLE reversed these harmful outcomes by downregulating inflammatory gene expression to values close to the control (untreated macrophages).

The LX-2 cell line was used as an in vitro model to evaluate the eventually hepatic anti-inflammatory and antifibrotic activity of MLE. In LX-2 cells treated with LPS, the expression of proinflammatory cytokines genes, IL-1β, IL-6, and COX-2 was evaluated. Figure 5 shows that, in human stellate cells treated with LPS, there was an increase in proinflammatory genes compared to the control group. The cotreatment with MLE reversed the previous trend, downregulating gene expression levels. Furthermore, the expression of fibrotic markers alpha smooth muscle actin (α-SMA), transforming growth factor beta (TGFβ), and collagen type I alpha 1 (COL1A1) was evaluated. LPS treatment strongly induced the expression of fibrotic genes, whereas MLE was able to reduce gene expression levels close to control group values. Lastly, we investigated heme oxygenase 1 (HO1) gene expression, a cellular oxidative stress marker [22,23]. We found an increase in HO1 expression following LPS treatment, while cotreatment with MLE was able to counteract this effect.

## 4. Discussion

Nutrition represents the gold standard for preventive health because it is based on a balanced composition in which the primary energy elements (carbohydrates, lipids, and proteins) and other constituents (vitamins and amino acids), combined with a series of phytochemicals, with high biological activity, such as polyphenols, are able to guarantee greater survival and longevity. The inclusion in the diet of bioactive elements favors the maintenance of a stationary dynamic equilibrium between the environment and the organism, through its metabolic machinery that processes them and allows the disposal of waste in a correct way. Recent scientific evidence suggests a growing interest in finding bioactive compounds obtained by natural products to be used in the formulation of food supplements and medicines.

In this context, fruits play an important role in the regular diet thanks to their several compounds with properties able to contrast aging and the related oxidative stress. The investigation of beneficial activities exerted by natural biomolecules is further corroborated by the avant-garde perspective of their practical application in different circumstances related to aging and its unavoidable consequences associated with cell damage, diabetes, inflammation, or fibrotic accumulation [24,25]. Bioactive compounds are secondary metabolites synthesized by plants in response to stressful conditions. Moreover, a good correlation between antioxidant performance and polyphenolics of the leaf of specific fruits was observed. Among these, mango leaves seem to be a suitable example, thanks to their wide range of bioactive compounds.

It has long been known that the accumulation of damage from free radicals (particularly RONS) in cells and tissues over the course of life is responsible for aging [26]. The consequences of oxidative stress have raised many theories that try to explain the phenomenon of aging; one of these is based on the idea that aging derives from the accumulation of deleterious effects caused by free radicals [27]. This would explain the increased ROS production found in aged tissues [28], which is considered the main cause of aging [29]. Further studies have also reported that the increase in oxidative damage is responsible for damage at level of lipids, nucleic acids, proteins, and carbohydrates, causing cellular dysfunction and making the body more susceptible to the development of related diseases [26].

Accordingly, numerous studies have focused on the ability of *Mangifera indica* L. content to constitute a valid resource in the restraint of oxidative and inflammatory events. In particular, evidence suggests that extracts obtained from different parts of *Mangifera indica* represent a useful source for maintaining a good health state, exerting a protective activity toward damage related to oxidative stress by improving outcomes for diseases such as respiratory diseases, atherosclerosis, cancer (e.g., breast, colon, neurons, skin, and neck), obesity, and diabetes [30,31,32].

Recent studies have largely debated on inexpensive and environmentally sustainable extraction methods of bioactive compounds from plant materials that minimize the use of solvents; especially for the use of plant extracts as food supplements/additives, extraction with food-grade solvents has been considered as the preferred condition for obtaining acceptable yields while safeguarding the precious bioactive features of plant components [11].

In our study, we evaluated the extract of leaves of *Mangifera indica* L. cultivated in Sicily (Italy), showing that MLE exhibited high concentrations of benzophenones, xanthones, and polyphenols. In particular, we found that the most concentrated compounds were iriflofenone 3-C-glucoside, mangiferin, and flavonols, mainly derived from quercetin. In particular, the biological benefits of these substances in many diseases are still being extensively evaluated by the scientific literature [33,34,35]. Furthermore, from the perspective of a good health state and the optimal clinical status of many different pathologies, it is essential to evaluate the substantial role played by microbial comorbidities. In this regard, we evaluated the antibacterial activity of MLE toward a wide group of bacterial strains. In particular, MLE showed a variable antibacterial activity, mainly toward *St. hemolyticus*, *St. aureus*, *Ps. fluorescens*, and *B. cereus* target strains. Moreover, our results showed an appreciable efficacy in reducing the growth of *B. subtilis*, *B. cereus*, *Ps. fluorescens*, *St. haemolyticus*, and *St. aureus*. Our results are consistent with those of Olasehinde et al. (2018) who demonstrated a powerful antibacterial activity of *M. indica* aqueous and ethanolic extracts against *St. aureus* and *Ps. aeruginosa* [15]. The results obtained by our investigation were further corroborated by the antimicrobial assay with the pure mangiferin solution at the same concentration of quantified mangiferin in our extract (2.02 mg/mL), used as the reference standard solution of MLE (see Table 1), which did not show any antimicrobial activity against the abovementioned target pathogens. Effectively, our data are inconsistent with the results of Singh et al. (2012), who demonstrated that mangiferin solution and its derivatives inhibited the growth of both bacterial species (*B. pumilus*, *B. cereus*, and *Salmonella* Virchow) and fungal species (*Thermoascus auranticus* and *Aspergillus flavus*). In addition, Singh et al. reported that mangiferin solutions with antimicrobial activity fluctuated in the concentration range of 15% to 30% [36]. Therefore, the antibacterial activity of MLE seems not to be directly correlated with the amount of mangiferin found in the extract but is probably due to the synergism of active compounds which constitute the total bioactive content of *Mangifera indica* L. Effectively, the concept of synergism among bioactive compounds is widely shared by the scientific literature, as suggested by Scavo et al. [37], regarding the antimicrobial activity of cultivated cardoon (*Cynara cardunculus* L. var. *altilis* DC.) leaf extracts. Furthermore, our data showed that MLE, due to its high concentration of benzophenones, xanthones, and polyphenols, exhibits in vitro inhibition of α-glucosidase, a glycoside hydrolase which hydrolyzes the terminal, nonreducing 1→4-linked α-d-glucose residues releasing glucose into the blood, resulting in postprandial hyperglycemia [38]. In this regard, Lawal et al. (2022) reported mixed competitive inhibition from *Camellia Sinensis* aqueous extracts, rich in flavonoids and catechins, against α-glucosidase [39]. In particular, many studies have confirmed this inhibitory effect exerted by several natural extracts. On this basis, several phenolic compounds, such as xanthones, contained in tea and wine, and recently discovered in black carrot juice, have shown a significant inhibition of this enzyme. Moreover, Kulkarni and Rathod (2018) [40] specifically evaluated the potential of mango leaf extract as inhibitor of α-glucosidase, speculating that the leaf extract and mangiferin could play an important role in the therapy of diabetes mellitus. In the perspective of pathologies strictly related to aging, such as diabetes, a recent medical challenge [41], data obtained showing the inhibitory activity of MLE toward glucosidase represent an interesting starting point for the potential application of mango extracts as an adjuvant for the treatment of this complex metabolic condition. In addition, the chemical, antioxidant, and antibacterial profile of MLE prompted us to further evaluate the anti-inflammatory ability of the red leaf extract on cellular in vitro models. In this regard, inflammation is a common biological condition representing the cell’s answer to different stimuli, often due to human pathogens such as viruses or bacteria, but also associated with the chronicization of different disorders including diabetes, cancer, cardiovascular diseases, obesity, and autoimmune diseases [42]. In order to obtain a reliable human inflammation model, we evaluated the expression of genes related to inflammation and fibrosis in differentiated macrophages (U937) and hepatic stellate cells (LX2) [43,44]. In particular, we used the inflammation induced by LPS as for a representation of stress damage and macrophage activation, and we analyzed the effect of LPS pretreatment with and without MLE treatment. MLE treatment was able to significantly counteract the inflammatory action of LPS, reducing the gene expression of proinflammatory enzymes and cytokines, such as IL-1β, IL-6, TNFα, and COX2. Furthermore, our data revealed the hepatic anti-inflammatory and antioxidant activity of MLE, resulting from decreased gene expression of cytokines IL-1β, IL-6, and COX2 following MLE cotreatment with LPS. Moreover, hepatic stress was further evaluated through the evaluation of fibrotic biomarkers related to fibrotic accumulation and extracellular matrix (ECM) disorders including α-SMA, TGFβ, and COL1A1 [45,46]. The gene expression of these markers was upregulated after LPS treatment in LX2 cells [22,47] and significantly decreased following MLE addition in the same cell group. Activation of stellate hepatic cells represents an event related to the early stages of liver fibrosis development; in fact, once activated, hepatic cells show a change in phenotype to myofibroblast-like cells with augmented production of ECM components, with the induced recruitment of inflammatory cells, leading to progressive fibrosis, liver failure, and progression from fatty liver to nonalcoholic steatohepatitis (NASH) [48,49]. Several studies have shown that hepatic fibrosis may be considered as a hepatic manifestation of metabolic syndrome, obesity, and diabetes [50,51,52].

Although these are preliminary analyses, our results suggest promising properties of MLE, and we will continue to investigate its effects against aging.

## 5. Conclusions

In the present study, we reported the analysis of some of the biological effects produced by bioactive compounds contained in MLE. In particular, the first aim was to select and evaluate a powerful mix of phytochemical components, selecting red leaves of *Mangifera indica* L. of the Kensington pride cultivar. Once the main bioactive molecules were quantified, it was important to evaluate the antioxidant activity due to the rich polyphenol derivatives offered by our extract. Once a satisfactory chemical profile of MLE was obtained, we studied its biochemical properties, concentrating on its antibacterial and antioxidant activities, as well as focusing on an inflammatory in vitro model offered by differentiated macrophages and hepatic stellate cells. The collected data revealed encouraging preliminary results, highlighting the importance of MLE in potential future applications due to its supportive properties in the treatment of metabolic syndrome-related disfunctions and aging-related disorders.

## Figures and Tables

**Figure 1 antioxidants-11-01313-f001:**
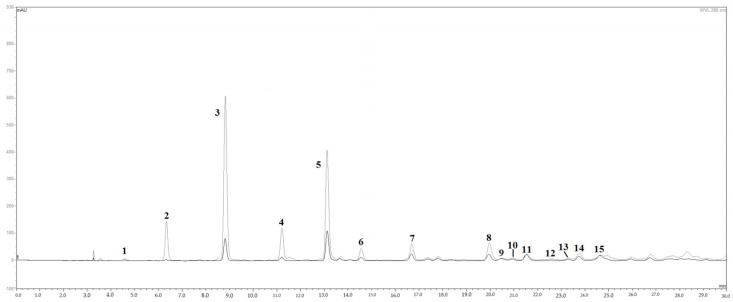
HPLC/DAD chromatograms of *M. indica* leaf extract. Fifteen peaks were tentatively identified; the six peaks over 15 (peaks 2, 3, 4, 6, 7, and 8) correspond to maclurin and iriflophenone derivatives, with iriflophenone 3-C-glucoside corresponding to peak #3, while mangiferin, is represented by peak 5. Peaks 11, 14, and 15 correspond to gallic acid (peak 1) and its derivatives with a hexose, plausibly glucose. Quercetin derivatives (peaks 9, 10, 12, and 13), are the only subclass of flavonoids found.

**Figure 2 antioxidants-11-01313-f002:**
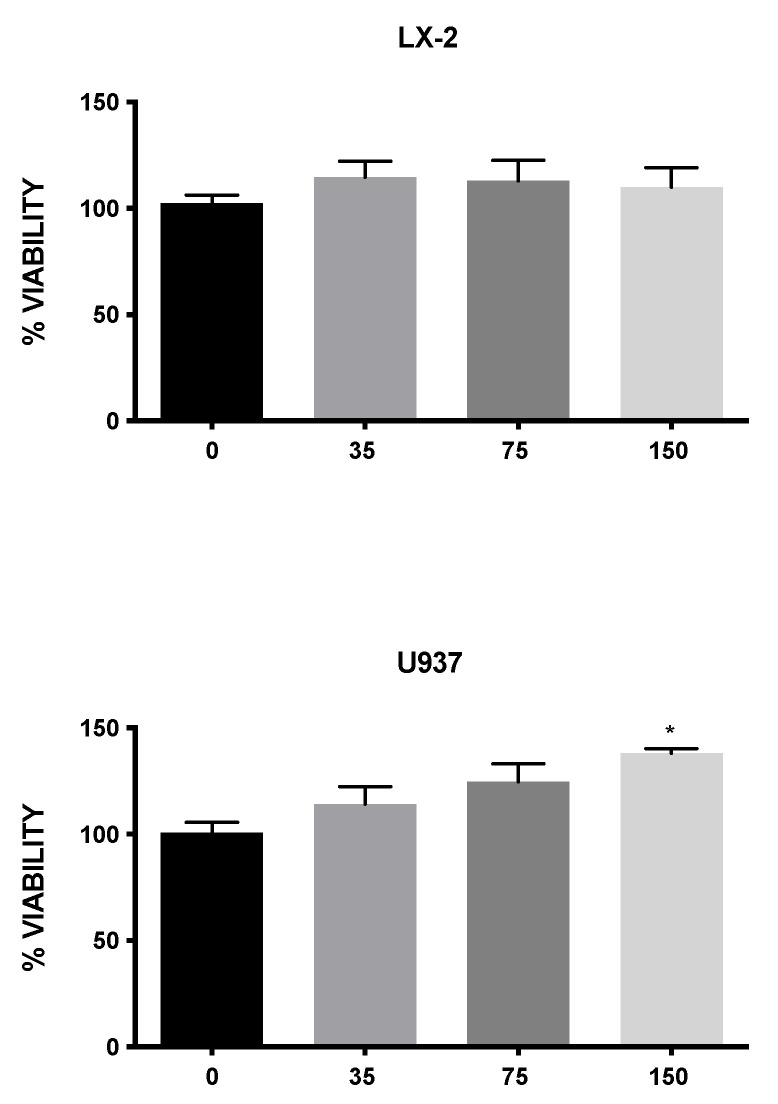
The viability assay of LX-2 and U937 cells according to bromide 3-(4,5-dimethylthiazol-2-yl)-2,5-diphenyltetrazolium (MTT) assay after treatment for 24 h with 0, 35, 75, and 150 μg/mL of MLE. Bars represent the mean ± SEM. * *p* < 0.5 versus control (CTRL).

**Figure 3 antioxidants-11-01313-f003:**
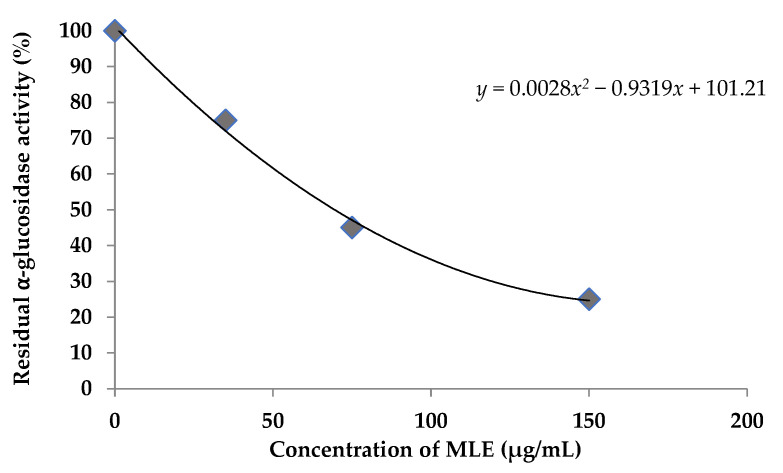
Inhibition of α-glucosidase by MLE. All assays were performed in triplicate.

**Figure 4 antioxidants-11-01313-f004:**
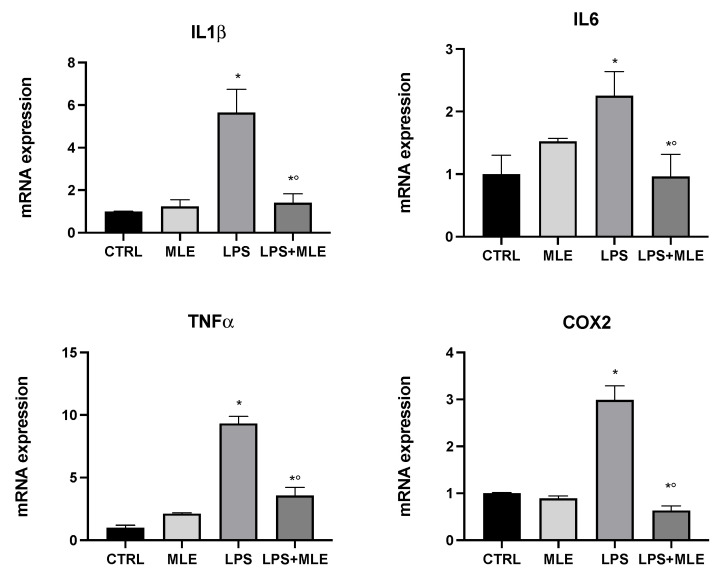
Gene expression in the macrophages of interleukin-1β (IL-1β), interleukin-6 (IL6), tumor necrosis factor-α (TNF-α), and cyclooxygenase-2 (COX-2) evaluated by RT-PCR. Bars represent the mean ± SEM of six independent experiments. * *p* < 0.05 versus undifferentiated cells; *° *p* < 0.05 versus differentiated cells.

**Figure 5 antioxidants-11-01313-f005:**
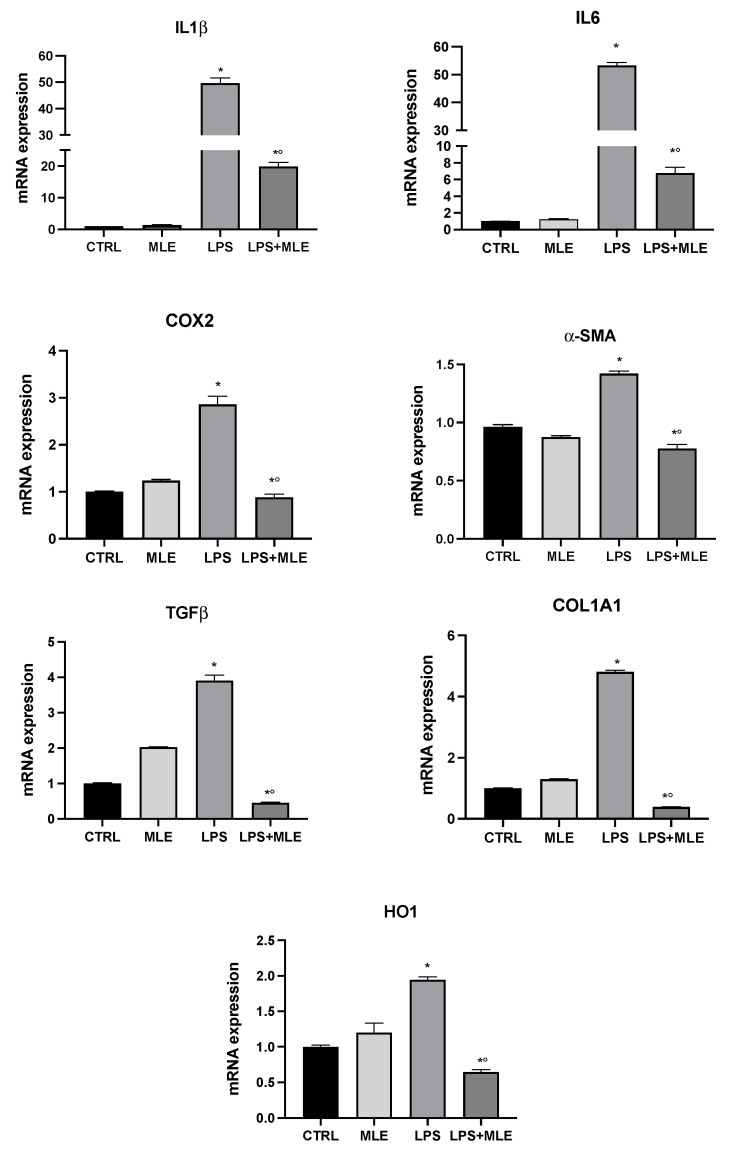
Gene expression in the hepatic stellate cells of interleukin-1β (IL-1β), interleukin-6 (IL6), tumor necrosis factor-α (TNF-α), cyclooxygenase-2 (COX-2), alpha smooth muscle actin (α-SMA), transforming growth factor beta (TGFβ), collagen type I alpha 1 (COL1A1), and heme oxygenase 1 (HO1) evaluated by RT-PCR. Bars represent the mean ± SEM of six independent experiments. * *p* < 0.05 versus undifferentiated cells; *° *p* < 0.05 versus differentiated cells.

**Table 1 antioxidants-11-01313-t001:** Primers used in this study.

Gene	Gene Accession Number	Forward Primer	Reverse Primer
α -SMA	NM_001141945.2	TCGCATCAAGGCCCAAGAAA	GGATTCCCGTCTTAGTCCCG
COX-2	NM_000963.4	CTGGCGCTCAGCCATACAG	CGCACTTATACTGGTCAAATCCC
COL1A1	NM_000088.4	CTGGCCTCCCTGGAATGAAG	GGCAGCACCAGTAGCACC
GAPDH	NM_002046.7	TTCTTTTGCGTCGCCAGCC	CTTCCCGTTCTCAGCCTTGAC
HO-1	NM_002133.3	GTTGGGGTGGTTTTTGAGCC	TTAGACCAAGGCCACAGTGC
IL-1β	NM_000576.3	ATGATGGCTTATTACAGTGGCAA	GTCGGAGATTCGTAGCTGGA
IL-6	NM_000600.5	CCACCGGGAACGAAAGAGAA	GAGAAGGCAACTGGACCGAA
TGF- β	NM_000660.7	GGAAATTGAGGGCTTTCGCC	CCGGTAGTGAACCCGTTGAT
TNF-α	NM_000594.4	GCAACAAGACCACCACTTCG	GATCAAAGCTGTAGGCCCCA

**Table 2 antioxidants-11-01313-t002:** Peak list and quantitative data of metabolites from *M. indica* leaf extract.

Peak	Rt, min ^a^	Compound Identification	Biochemical Class	mg/mL
1	4.62	Gallic acid ^b^	Phenolic acid	0.028
2	6.47	Maclurin 3-*C*-glucoside	Benzophenones/xanthones	0.608
3	8.92	Iriflophenone 3-*C*-glucoside	Benzophenones/xanthones	4.218
4	11.31	Iriflophenone 3-C-(2-*O*-galloyl) glucoside	Benzophenones/xanthones	1.089
5	13.15	Mangiferin ^b^	Benzophenones/xanthones	2.023
6	14.66	Iriflophenone derivative isomer 1 ^c^	Benzophenones/xanthones	0.368
7	16.77	Iriflophenone 3-*C*-(2,6 di-*O*-galloyl) glucoside	Benzophenones/xanthones	0.960
8	20.07	Iriflophenone derivative isomer 2 ^c^	Benzophenones/xanthones	1.082
9	20.55	Rutin ^b^	Flavonols	0.056
10	21.04	Quercetin 3-*O*-glucoside	Flavonols	0.030
11	21.62	Galloyl hexose isomer 1 ^c^	Phenolic acid	0.212
12	22.59	Quercetin pentoside isomer 1 ^c^	Flavonols	0.012
13	23.41	Quercetin pentoside isomer 2 ^c^	Flavonols	0.019
14	23.89	Galloyl hexose isomer 2 ^c^	Phenolic acid	0.269
15	24.82	Galloyl hexose derivative ^c^	Phenolic acid	0.206
		Total gallic acid derivatives		0.715
		Total benzophenones/xanthones		10.350
		Total flavonols		0.118
		Total polyphenols		11.184

^a^ Mean of three replicates; ^b^ co-injection with pure analytical standard; ^c^ correct isomer not identified.

**Table 3 antioxidants-11-01313-t003:** Antioxidant activity of mango leaf extract and mangiferin standard.

Sample	RSA (%)
Leaf Extract	90.19 ± 0.31 ns
Mangiferin (2.02 mg/mL)	87.80 ± 1.86 ns

Data are presented as the mean ± standard deviation. Radical-scavenging activity (RSA) values followed by a letter are significantly different according to Fisher’s least significant difference test (*p* ≤ 0.05); ns = not significant.

**Table 4 antioxidants-11-01313-t004:** Antibacterial activity of mango leaf extract (MLE) and mangiferin standard.

	Antibacterial Activity	
	Inhibition Zone (cm)	
Microorganism	MLE	Mangiferin (2.02 mg/mL)
*B. cereus*	0.43 ± 0.06 c	0.00 ± 0.00 ns
*B. subtilis*	0.23 ± 0.06 d	0.00 ± 0.00 ns
*Ps. fluorescens*	0.70 ± 0.00 a	0.00 ± 0.00 ns
*St. aureus*	0.43 ± 0.06 c	0.00 ± 0.00 ns
*St. haemolyticus*	0.50 ± 0.00 b	0.00 ± 0.00 ns

Data are presented as the mean ± standard deviation. In each column, values followed by a different letter, within the same column, are significantly different according to Fisher’s least significant difference test (*p* ≤ 0.05); ns = not significant.

## Data Availability

The data presented in this study are available in the article.

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
