# Peer review of "Mangifera indica L. Leaves as a Potential Food Source of Phenolic Compounds with Biological Activity"

_antioxidants, 2022, doi:10.3390/antiox11071313_

Round 1
Reviewer 1 Report
Article
Mangifera Indica L. Leaves as a Potential Food Source of Phenolic Compounds with Biological Activity
A brief summary
The idea of using plant raw materials that are waste in the normal production process is not new, but it is worthy of attention, especially today. The paper seems to be interesting and quite valuable. The research is well designed and performed but documentation and description needs some fixes and additions. Nevertheless, the article is clearly written and the presentation style is appropriate for a scientific journal.
Broad comments
1. In the discussion, the Authors cite the trend of using "inexpensive and environmentally-sustainable 370 extraction methods" and "food-grade solvents" (lines 370, 373). How would the Authors define method and solvent used in this work?
2. Have the chromatographic methods used been validated according to generally accepted principles?
3. Typos should be corrected and descriptions should be standardised throughout the work - in the text and the captions under the illustrations. (eg. Table 2 and lines 92 – 96 commas instead of dots and R2 - 2 as superscript, Camellia Sinensis).
4. Keywords - instead of "Magniflora indica L." (it is already in the title) I suggest simply "mango leaves".
Specific comments
Line 65 and 117: Was it really once bought from Sigma in Italy and once from the USA? Also, once it is "Sigma Aldrich" and once it is "Sigma-Aldrich". And of course the reference in line 117 is to methanol and not DPPH?
Line 79: The entire first sentence is unnecessary.
Lines 92 - 96: Please replace commas with dots as decimal separators and R2 - 2 as superscript
Teble 2: Last column - in which units is the content of the determinated compounds given? This is probably one of the most important parameters addressed in this work! Please replace commas with dots as decimal separators.
Table 3: If no differences are found, it is better to write "ns" instead of letters with means - not significant. The character size of these letters should be harmonised with the other tables (see Table 4).
Figure 2, 3 and 4: Please standardise the notation of units - ug/ml or ug/mL.
Line 335: The phrase 'secondary components' may perhaps be somewhat confusing with 'secondary metabilites'. This is probably about something else.
Author Response
Response to reviewer 1
We would like to thank the reviewer for comments and suggestions in order to improve the paper. Please find the answers point by point below.
Comments and Suggestions for Authors
Reviewer 1
Article
Mangifera Indica L. Leaves as a Potential Food Source of Phenolic Compounds with Biological Activity
A brief summary
The idea of using plant raw materials that are waste in the normal production process is not new, but it is worthy of attention, especially today. The paper seems to be interesting and quite valuable. The research is well designed and performed but documentation and description needs some fixes and additions. Nevertheless, the article is clearly written and the presentation style is appropriate for a scientific journal.
Broad comments
- In the discussion, the Authors cite the trend of using "inexpensive and environmentally-sustainable 370 extraction methods" and "food-grade solvents" (lines 370, 373). How would the Authors define method and solvent used in this work?
Thank you for your comment. In this work, a very simple and time saving method was used for the extraction of bioactive compounds from their vegetable matrix. In fact, the solvent mixture used was 70% ethanol (food grade) in distilled water, and the time required for the extraction was 30 minutes only (paragraph 2.2)
- Have the chromatographic methods used been validated according to generally accepted principles?
Not at this stage of our research, which is still at a very preliminary level. We thank the reviewer for his/her hint, as we definitely should validate the method (in terms of LOD, LOQ, cutoff value, stability, selectivity and precision) when upgrading our work to the analysis of mango leaves containing drugs and matrices for QC purposes, as excellently suggested by Indrayanto in his recent Chapter “Validation of Chromatographic Methods of Analysis: Application for Drugs That Derived From Herbs” (Profiles of Drug Substances, Excipients and Related Methodology 2018, 43, 359-392). Method validation is not usually required for preliminary stage experiments.
- Typos should be corrected and descriptions should be standardised throughout the work - in the text and the captions under the illustrations. (eg. Table 2 and lines 92 – 96 commas instead of dots and R2 - 2 as superscript, Camellia Sinensis).
We standardized the paper and followed the reviewer's advice
- Keywords - instead of "Magniflora indica L." (it is already in the title) I suggest simply "mango leaves".
Thank you for your comment. As suggested we used mango leaves rather than the full name
Specific comments
Line 65 and 117: Was it really once bought from Sigma in Italy and once from the USA? Also, once it is "Sigma Aldrich" and once it is "Sigma-Aldrich". And of course the reference in line 117 is to methanol and not DPPH?
Thanks, we have corrected as you suggested.
Line 79: The entire first sentence is unnecessary.
Thanks, we have corrected as you suggested.
Lines 92 - 96: Please replace commas with dots as decimal separators and R2 - 2 as superscript
Thanks, we have corrected as you suggested.
Teble 2: Last column - in which units is the content of the determinated compounds given? This is probably one of the most important parameters addressed in this work! Please replace commas with dots as decimal separators.
mg/mL was added in the last column of table 2(thank you!) and commas were replaced by dots as suggested.
Table 3: If no differences are found, it is better to write "ns" instead of letters with means - not significant. The character size of these letters should be harmonised with the other tables (see Table 4).
Thanks, we have corrected as you suggested.
Figure 2, 3 and 4: Please standardise the notation of units - ug/ml or ug/mL.
Thanks, we have corrected as you suggested.
Line 335: The phrase 'secondary components' may perhaps be somewhat confusing with 'secondary metabilites'. This is probably about something else.
Thanks, we have corrected as you suggested.
Reviewer 2 Report
The paper is overall well written.
With effects of extract presented towards cellular effects, there is a lack of discussion on the availability and therefore relevance of these towards health/chronic disease.
Recommendation to carefully proof-read to remove English language errors.
Cell culture experiments not made clear, there is description of MTT assay in general and then incubations described for LX-2 and U937 cells, however, cytotox are shown for LX2 and BEAS-2B. Why was U937 not used, this is inconsistent. Clarify and potentially remove the cell line that is not referred to otherwise, unless there is a particular reason to leave in.
Improve figures, e.g. by removing the full descripton under each bar, the number for concentration is sufficient, e.g. 150 - the unit can be presented in legend. Also, the lowest concentration needs to be after control and not the highest. If control is 0, then label as such to make it consistent. What does it mean, in fig3, six independent experiments? cell culture expts are usually done in multiplicates in one passage (which is then repeated twice), this is not clear from description). Why are fig2 and 3 not combined?
re alpha-glucosidase, calculate IC50.
The inflammatory model is not clear. What exactly was assessed? It is in fact unusual (in particular in macrophages) to apply inflammatory stimulus for a few hours and then add the extract? given the dynamics of inflammatory response, the expectation for the rise in proinflammatory markers, in particular TNFa, is different. Referencing/further data on this required.
Adapt a-glucosidase figure to style of others.
fig5 & 6, possible to shorten label y axis? seems to long. Remove the titles with cell lines, this infromation should be presented in figure legend. Again, what are six independent experiments? is this n=1 in six passages or n=6 in one passage?
Table 2 - there is a heading missing for the last column in the table.
Table 3 - add concentration of leaf extract
Discussion, remove proper as first word, what is the definition of proper nutrition ,not appearing scientific expression.
rephrase in line 394 - don't perfectly fit to more scientific
format: apply italics for bacterial species throughout
L439 - properly reduced (and remove this word elsewhere)
Author Response
Comments and Suggestions for Authors
The paper is overall well written.
With effects of extract presented towards cellular effects, there is a lack of discussion on the availability and therefore relevance of these towards health/chronic disease.
Recommendation to carefully proof-read to remove English language errors.
Cell culture experiments not made clear, there is description of MTT assay in general and then incubations described for LX-2 and U937 cells, however, cytotox are shown for LX2 and BEAS-2B. Why was U937 not used, this is inconsistent. Clarify and potentially remove the cell line that is not referred to otherwise, unless there is a particular reason to leave in.
Thanks for the comment. As suggested by the reviewer, we performed new cytotoxicity experiments on the U937 cell line. We had previously used the beas-2 cell line because it is more susceptible than other cell lines. The paper is now more consistent. Thanks again.
Improve figures, e.g. by removing the full descripton under each bar, the number for concentration is sufficient, e.g. 150 - the unit can be presented in legend. Also, the lowest concentration needs to be after control and not the highest. If control is 0, then label as such to make it consistent. What does it mean, in fig3, six independent experiments? cell culture expts are usually done in multiplicates in one passage (which is then repeated twice), this is not clear from description). Why are fig2 and 3 not combined?
Thanks, we have corrected as you suggested.
re alpha-glucosidase, calculate IC50.
Thank you for comment, we added IC50 in the paper.
The IC50 value of enzymatic inhibition was calculated using the linear plot equation of inhibition curve.
The inflammatory model is not clear. What exactly was assessed? It is in fact unusual (in particular in macrophages) to apply inflammatory stimulus for a few hours and then add the extract? given the dynamics of inflammatory response, the expectation for the rise in proinflammatory markers, in particular TNFa, is different. Referencing/further data on this required.
Thank you for your comment. We add more information in methods to clarify dynamic of experiments. Our experiments are consistent with the papers cited below. Addition of the extract after treatment with LPS we wanted to investigate the possibility of mitigating the induced inflammation.
Facchin BM, Dos Reis GO, Vieira GN, Mohr ETB, da Rosa JS, Kretzer IF, Demarchi IG, Dalmarco EM. Inflammatory biomarkers on an LPS-induced RAW 264.7 cell model: a systematic review and meta-analysis. Inflamm Res. 2022 May 25. doi: 10.1007/s00011-022-01584-0. Epub ahead of print. PMID: 35612604.
Min HY, Kim H, Lee HJ, Yoon NY, Kim YK, Lee HY. Ethanol Extract of <i>Sargassum siliquastrum</i> Inhibits Lipopolysaccharide-Induced Nitric Oxide Generation by Downregulating the Nuclear Factor-Kappa B Signaling Pathway. Evid Based Complement Alternat Med. 2022 Jun 10;2022:6160010. doi: 10.1155/2022/6160010. PMID: 35722164; PMCID: PMC9205721.
Adapt a-glucosidase figure to style of others.
Thanks, we have corrected as you suggested.
fig5 & 6, possible to shorten label y axis? seems to long. Remove the titles with cell lines, this infromation should be presented in figure legend. Again, what are six independent experiments? is this n=1 in six passages or n=6 in one passage?
Thanks, we have corrected and clarify as you suggested.
Table 2 - there is a heading missing for the last column in the table.
Thanks, we have corrected as you suggested.
Table 3 - add concentration of leaf extract
Thanks, we have corrected as you suggested.
Discussion, remove proper as first word, what is the definition of proper nutrition ,not appearing scientific expression.
Thanks, we have corrected as you suggested.
rephrase in line 394 - don't perfectly fit to more scientific
Thanks, we have corrected as you suggested.
format: apply italics for bacterial species throughout
Thanks, we have corrected as you suggested.
L439 - properly reduced (and remove this word elsewhere)
Thanks, we have corrected as you suggested.